# Is Circulating Vitamin D Status Associated with the Risk of Venous Thromboembolism? A Meta-Analysis of Observational Studies

**DOI:** 10.3390/nu15051113

**Published:** 2023-02-23

**Authors:** Kuo-Chuan Hung, Sheng-Hsiang Yang, Chia-Yu Chang, Li-Kai Wang, Yao-Tsung Lin, Chia-Hung Yu, Min-Hsiang Chuang, Jen-Yin Chen

**Affiliations:** 1Department of Anesthesiology, Chi Mei Medical Center, Tainan 71004, Taiwan; 2Department of Hospital and Health Care Administration, College of Recreation and Health Management, Chia Nan University of Pharmacy and Science, Tainan 71710, Taiwan; 3School of Medicine, College of Medicine, National Sun Yat-Sen University, Kaohsiung 80424, Taiwan; 4Department of Neurology, Chi Mei Medical Center, Tainan 71004, Taiwan; 5The Center for General Education, Southern Taiwan University of Science and Technology, Tainan 71004, Taiwan; 6Department of Internal Medicine, Chi Mei Medical Center, Tainan 71004, Taiwan

**Keywords:** 25-hydroxyvitamin D, vitamin D deficiency, vitamin D insufficiency, venous thromboembolism, risk factor

## Abstract

Background: Although vitamin D is antithrombotic, associations between serum vitamin D status and the risk of venous thromboembolism (VTE) remain inconsistent. Methods: We searched the EMBASE, MEDLINE, Cochrane Library, and Google Scholar databases from inception to June 2022 to identify observational studies examining associations between vitamin D status and VTE risk in adults. The primary outcome presented as odds ratio (OR) or hazard ratio (HR) was the association of vitamin D levels with the risk of VTE. Secondary outcomes included the impacts of vitamin D status (i.e., deficiency or insufficiency), study design, and the presence of neurological diseases on the associations. Results: Pooled evidence from a meta-analysis of sixteen observational studies, including 47648 individuals published from 2013 to 2021, revealed a negative relationship between vitamin D levels and the risk of VTE either based on OR (1.74, 95% confidence interval (CI): 1.37 to 2.20, *p* < 0.00001; I^2^ = 31%, 14 studies, 16074 individuals) or HR (1.25, 95% CI: 1.07 to 1.46, *p* = 0.006; I^2^ = 0%, 3 studies, 37,564 individuals). This association remained significant in subgroup analyses of the study design and in the presence of neurological diseases. Compared to individuals with normal vitamin D status, an increased risk of VTE was noted in those with vitamin D deficiency (OR = 2.03, 95% CI: 1.33 to 3.11) but not with vitamin D insufficiency. Conclusions: This meta-analysis demonstrated a negative association between serum vitamin D status and the risk of VTE. Further studies are required to investigate the potential beneficial effect of vitamin D supplementation on the long-term risk of VTE.

## 1. Introduction

Venous thromboembolism (VTE), which is considered a serious public health problem due to reduced survival, high healthcare costs, and frequent recurrence, may result in severe complications [1]. A major theory for the pathogenesis of VTE is known as the Virchow triad, which comprises alterations in blood flow, vascular endothelial injury, and alterations in the constituents of blood [2]. The incidence of VTE varies substantially among different ethnic groups [3]. A cohort study using the California Discharge Data from 1996 revealed that the incidence of VTE per 100,000 was 104 in Caucasians, 141 in African-Americans, 55 in Hispanics, and 21 in Asians, indicating a higher incidence of VTE in African-Americans compared with a relatively low incidence among Asians [3]. The causes and risk factors of VTE can be inherited or acquired [4,5] and include advancing age, women, obesity, smoking, active cancer, trauma, fracture, immobility, some medications, and nutritional status [1,6,7]. Identification of the underlying risk factors, which are modifiable or non-modifiable [8], can facilitate early diagnosis. In particular, targeting the modifiable risk factors with appropriate interventions may help prevent thrombotic events.

Vitamin D, a secosteroid hormone, is well-known for the maintenance of calcium and bone homeostasis [9]. It has been found to exert extraskeletal pleiotropic effects on the immune, neurological, and cardiovascular systems [10,11,12]. Vitamin D can also prevent vascular endothelial injury [13] through its anti-inflammatory [14] and anti-oxidative [15] properties. Furthermore, vitamin D is antithrombotic because of its inhibition of the coagulation pathway [16,17]. Therefore, vitamin D deficiency may theoretically lead to a prothrombotic state that may contribute to the development of VTE [13]. Indeed, vitamin D deficiency, which is a modifiable nutritional status, has been hypothesized as a possible risk factor for VTE and gained attention in recent years [7,18]. However, some studies have shown no relationship between serum vitamin D status and the incidence of VTE [19,20]. Variations in sample sizes and study designs may have contributed to these conflicting results, which warranted a systematic analysis to generate reliable evidence.

To date, one previous meta-analysis focusing on seven studies demonstrated a significant association between a low vitamin D level and an increased risk of VTE [21]. Nevertheless, a potential dose-dependent effect and some concern over data synthesis [i.e., the use of hazard ratio (HR) as a substitute for odds ratio (OR)] [21] may limit the application of its findings. In addition, the lack of association between vitamin D status and VTE in that meta-analysis focusing on case-control studies [21] suggested impaired evidence on this issue. The current meta-analysis attempted to clarify the association between the risk of VTE and vitamin D status (i.e., primary outcome) to address these issues. Furthermore, we extended our investigations into the potential impacts of vitamin D status (i.e., deficiency or insufficiency), study designs [21], and the presence of neurological diseases [22] in secondary analyses.

## 2. Materials and Methods

The current meta-analysis was conducted in accordance with the Preferred Reporting Items for Systematic Reviews and Meta-analyses (PRISMA) checklist [23] and registered in the International Prospective Register of Systematic Reviews (PROSPERO, https://www.crd.york.ac.uk/prospero/, accessed on 4 February 2021, registration no.: CRD42021228606).

### 2.1. Sources of Data and Search Strategies

Potentially eligible studies were retrieved from the databases of EMBASE, MEDLINE, Cochrane Library, as well as Google Scholar from inception to June 10, 2022, by using different keyword and MeSH term combinations, including (“hypovitaminosis D” or “Vitamin D” or “25-hydroxyvitamin D” or “Vitamin D Deficien*” or “25OHD” or “vitamin D2” or “vitamin D3” or “25(OH)D” or “Hydroxycholecalciferols”) and (“Deep vein thrombosis*” or “DVT” or “Thromboembolism” or “Venous Thrombos*” or “Deep-vein thrombosis*” or “Venous Thromboembolism” or “Pulmonary embolism*” or “Pulmonary Thromboembolism” or “Thromboembolic” or “PE”). We posed no restrictions on language, sample size, year of publication, and distribution of genders. Detailed descriptions about the use of search terms (i.e., Medline), as well as the method for article retrieval, are shown in Appendix A. The included references of related systematic review articles and the eligible studies were also scrutinized to identify potentially eligible studies to be included in the current investigation.

### 2.2. Inclusion and Exclusion Criteria

Studies including case-control, retrospective, and population-based study designs were considered eligible if they examined the correlation between vitamin D level and risk of VTE in adults without restriction to the duration of follow-up. For studies recruiting mixed populations (i.e., adults and adolescents), we included all participants in our analysis if the mean or median age of the study population was more than 30 years on the assumption of their inclusion of at least 50% of adult participants. We excluded (1) studies that were case reports, non-human studies, conference proceedings, reviews, and letters; (2) those that did not provide data for the calculation of individual OR or HR with corresponding 95% confidence intervals (CI); and (3) those in which thrombosis in the low extremities or pulmonary vein was not reported (e.g., studies focusing on cerebral venous sinus thrombosis). Two independent authors surveyed the titles together with the abstracts of the acquired articles. All conflicts were settled through discussion. Articles that passed the title and abstract screening were selected for a full-text review and subsequently chosen for the final analysis after agreements between authors.

### 2.3. Data Extraction and Risk of Bias Assessment

The following details were retrieved from each study: author information, year of publication, age and gender of participants, characteristics of the study population, sample size, event rate of VTE, country, study design (e.g., case-control study), methods for quantification of serum 25-hydroxyvitamin D [25(OH)D], and circulating vitamin D levels. We contacted the authors of the eligible articles for missing data if necessary.

The Newcastle-Ottawa Scale (NOS), which scores each study based on three domains (i.e., study group selection, group comparability, and outcome of interest ascertainment), was used to assess the quality of our included observational studies as previously described [24]. The total maximum score is nine. A study with a score of more than seven was regarded as having a low risk of bias. Using a standard data-extraction form, two authors separately extracted relevant data and assessed the risk of bias. The results of the evaluation were verified by a third author, who was also responsible for resolving any discrepancies between the two authors.

### 2.4. Outcomes and Definitions

Because serum 25(OH)D is often used to assess vitamin D status in clinical practice [25], our primary outcome was to determine the correlation of a low circulating vitamin D status according to serum 25(OH)D level with the risk of VTE based on a binary comparison between participants with a serum 25(OH)D level above the cutoff concentration and those with a level below the cutoff value, which was defined according to each study [i.e., 20 or 30 ng/mL]. The diagnosis of VTE was in accordance with that of each study. To further elucidate the presence of a dose-dependent association between vitamin D level and the risk of VTE, subgroup analysis was conducted on patients based on categorizing their serum 25(OH)D levels into different ranges (i.e., <20 ng/mL vs. ≥30 ng/mL; 20–30 ng/mL vs. ≥30 ng/mL). In the current meta-analysis, vitamin D deficiency and insufficiency were defined as serum 25(OH)D levels of <20 ng/mL and 20–30 ng/mL, respectively [9,26]. Albeit somewhat arbitrary, a variation of 10% in 25(OH)D level was considered acceptable for the current study (e.g., a level of 18–28 ng/mL to be categorized as vitamin D insufficiency). In addition, a previous meta-analysis reported that study design (i.e., case-control) might have an impact on the correlation between vitamin D status and the risk of VTE [21]. Therefore, subgroup analyses were performed based on the study design (i.e., non-case-control vs. case-control) and the presence of neurological diseases [22] as secondary outcomes in the current study.

### 2.5. Data Synthesis and Analysis

All statistical analyses of this study were performed using the Cochrane Review Manager (RevMan 5.3; Copenhagen: The Nordic Cochrane Centre, The Cochrane Collaboration, 2014). For the primary analysis, an overall OR with 95% CI was obtained through the calculation of the overall effect size from the reported raw data of event counts using a random-effects model. If a study also provided details regarding HR, the pooled effect size was reported if more than two studies were available. Heterogeneity in effect size was evaluated using I-squared statistics with significant heterogeneity being defined as an I^2^ above 50% [27], for which leave-one-out sensitivity analysis was performed to examine the reliability of the available evidence. Potential publication bias was identified by visual inspection of a funnel plot and Egger’s test when 10 or more trials shared an outcome. A probability value (*p*) less than 0.05 was deemed statistically significant.

## 3. Results

### 3.1. Study Selection, Characteristics of Studies, and Risk of Bias Assessment

The database screening process is shown in Figure 1. A literature search identified 1077 citations in total. Following the removal of duplicates (n = 126), the titles and abstracts of the remaining 951 articles were examined. Further removal of 914 studies following examination of their titles and abstracts gave 37 articles for full-text review. Finally, 16 observational studies that fulfilled the inclusion criteria, published from 2013 to 2021, involving 47648 individuals, were meta-analyzed for the present investigation [7,18,19,20,28,29,30,31,32,33,34,35,36,37,38,39] (Figure 1).

The characteristics of the included studies are summarized in Table 1. The age of the individuals varied from 45 to 81.7 years, with male prevalence ranging from 24.3% to 72%. The association of vitamin D status with the risk of VTE was assessed in a wide range of patients, including elderly individuals (one study) [28], the general population (three studies) [7,19,20], those with COVID-19 (two studies) [29,37], those with VTE (four studies) [18,30,32,35], those undergoing surgery (two studies) [33,34], and those with neurological impairment (e.g., stroke or spinal cord injury) (four studies) [31,36,38,39]. The size of the study population ranged from 50 to 18791. Information about the three categories of the 25(OH)D level, namely <20, 20–30, and >30 ng/mL, was available in nine studies [18,28,29,30,31,35,36,38,39], while one study divided the 25(OH)D levels as <18, 18–28, >28 ng/mL [19]. On the other hand, four other studies only defined the serum 25(OH)D level as low or high (<20 vs. >20 ng/mL, respectively) without providing detailed information about serum vitamin D concentration [32,33,34,37]. Three studies were available for the calculation of HR [7,19,20]. The results of the study quality assessment are shown in Table 1. Three studies were considered to be of low quality [7,19,20], while the other 13 studies were judged to be having a low risk of bias based on the NOS scores that ranged from 7 to 9 (Table 1).

### 3.2. Outcomes

The outcomes of the current meta-analysis were derived from 16 observational studies. Among these studies, 13 reported OR as a measure of the association between serum vitamin D status and the risk of VTE, while two presented the cumulative incidence of VTE at a specific time point as HR. One study described both parameters (i.e., HR and OR) [19].

#### 3.2.1. Association of Vitamin D Status with the Overall Risk of VTE

##### Association of Low Vitamin D with the Risk of VTE Estimated with Odds Ratio

A low vitamin D level was defined as 25(OH)D level <20 ng/mL and <30 ng/mL in four [32,33,34,37] and ten studies [18,19,28,29,30,31,35,36,38,39], respectively. A pooled analysis of the association between vitamin D status (i.e., low vs. normal) and the overall risk of VTE demonstrated that a low vitamin D level was significantly associated with an increased risk of VTE when analyzed with a random-effects model (OR = 1.74, 95% CI: 1.37 to 2.20, *p* < 0.00001; I^2^ = 31%, 14 studies, 16,074 individuals) (Figure 2) [18,19,28,29,30,31,32,33,34,35,36,37,38,39]. The funnel plot did not reveal evidence of significant asymmetry (Figure 3), suggesting a low risk of publication bias. Meta-analysis of the four studies [28,33,37,39] that provided the adjusted OR for analyzing the impact of a low vitamin D level on the risk of VTE also showed a positive association between the former and the latter (adjust OR: 1.71, 95% CI:1.24 to 2.38, *p* = 0.001, I^2^ = 45%, 7503 patients) (Figure not shown).

##### Association of Low Vitamin D with the Risk of VTE Based on a Time-to-Event Analysis

Three studies involving 37,564 individuals provided information for the calculation of HR. The pooled result also revealed a negative association of vitamin D status with the risk of VTE (HR = 1.25, 95% CI: 1.07 to 1.46, *p* = 0.006; I^2^ = 0%, 3 studies, 37,564 individuals) (Figure 4) [7,19,20].

#### 3.2.2. Subgroup Analyses

Eight studies were available to analyze the dose-dependent association of vitamin D level with the risk of VTE [18,19,28,29,30,35,38,39]. The results showed that the risk of VTE in individuals with vitamin D deficiency [i.e., 25(OH)D level <20 ng/mL] was higher than that in those with 25(OH)D level ≥30 ng/mL) (OR = 2.03, 95% CI: 1.33 to 3.11, *p* = 0.001; I^2^ = 65%, eight studies, 4564 individuals) (Figure 5) [18,19,28,29,30,35,38,39]. Omitting certain trials had no significant effect on the results of the sensitivity analysis. Merged results from eight studies revealed no significant difference in the risk of VTE in individuals with vitamin D insufficiency [i.e., 25(OH)D level of 20–30 ng/mL] compared to those with 25(OH)D level ≥30 ng/mL (OR = 1.05, 95% CI: 0.63 to 1.76, *p* = 0.84; I^2^ = 66%, eight studies, 5325 individuals) (Figure 6). Sensitivity analysis demonstrated that a 25(OH)D level of 20–30 ng/mL was associated with an increased risk of VTE when one study was removed [28].

As study design has been reported to have an impact on the correlation between circulating vitamin D levels and the risk of VTE [21], we assessed the potential difference in the risk of VTE according to study design (i.e., non-case-control vs. case-control) using subgroup analysis. Although a negative association was noted between serum vitamin D level and the risk of VTE when “non-case-control” (OR = 1.60, 95% CI: 1.19 to 2.16, *p* = 0.002) and “case-control” (OR = 1.93, 95% CI: 1.29 to 2.88, *p* =0.001) studies were compared with their respective controls, there was no significant influence of research design on study outcome of subgroup analysis (*p* = 0.47) (Figure 7).

The impact of neurological diseases on the overall risk of VTE in participants with or without a low vitamin D level is shown in Figure 8. Subgroup analysis identified a low vitamin D status as a risk factor for VTE in patients with (OR = 2.12, 95% CI: 1.51 to 2.99, *p* < 0.0001) or without (OR = 1.58, 95% CI: 1.19 to 2.11, *p* = 0.002) neurological diseases. On the other hand, subgroup analysis comparing the two subgroups showed no significant difference (*p* = 0.2).

## 4. Discussion

This updated meta-analysis of sixteen observational studies enrolling 47,648 individuals revealed a negative relationship between vitamin D status and the risk of VTE. The low heterogeneity (I^2^ = 31%) and the absence of significant publication bias in the current study highlighted the robustness of the derived evidence. This meta-analysis demonstrated that vitamin D deficiency may be linked to an increased risk of VTE. Subgroup analysis further revealed no significant impact of study designs (e.g., case-control vs. non-case-control) and the presence of neurological diseases on the risk of VTE.

The finding of our updated study showing an association between vitamin D deficiency and an increased risk of VTE is consistent with that of a previous meta-analysis [21]. According to the Virchow triad of thrombus formation, namely endothelial injury, hypercoagulability, and venous stasis [2], the antithrombotic effects of vitamin D may be explained by the following mechanisms. First, there is evidence showing an anticoagulant effect of vitamin D through preventing endothelial dysfunction and suppressing inflammation [13]. The active form of vitamin D, 1,25(OH)2D, exerts anti-inflammatory effects by inducing the expression of interleukin (IL)-10 (an anti-inflammatory cytokine) and IL-10 receptor [40]. Besides, a cohort study on a normal population demonstrated an inverse association of plasma 25(OH)D levels with the concentrations of asymmetric dimethylarginine (a mediator of endothelial dysfunction) and high-sensitivity C-reactive protein (a marker of inflammation), both of which contribute to the onset and progression of thrombosis [41]. Second, accumulated evidence showed that vitamin D prevents blood clot formation by a direct or indirect blockade of the coagulation pathway. It was proposed that the upregulation of tissue factor (a key trigger of coagulation) and downregulation of thrombomodulin (an anticoagulant glycoprotein) in endothelial and monocytic cells are the main thrombogenic mechanisms [13]. Importantly, endothelial expression of tissue factor is regulated by nuclear factor-kappa B (NF-kB) in circulating mononuclear cells, which is a central mediator of inflammation related to thrombotic formation [16]. It was shown that 1,25(OH)2D inhibits endothelial tissue factor expression via direct inhibition of NF-kB activation [14]. Another study demonstrated that tumor necrosis factor (TNF) stimulates mRNA and protein expressions of tissue factor as well as inhibits the expressions of mRNA and protein of thrombomodulin in cultured endothelial and monocytic cells. Intriguingly, 1,25(OH)2D can reduce gene transcription of the tissue factor and enhance gene transcription of thrombomodulin to counteract the effects of TNF on coagulation [17]. Third, suppression of fibrinolysis is critical for the development of venous stasis-induced thrombus. It is proposed that elevated plasminogen activator inhibitor-1 (PAI-1) could suppress fibrinolysis and increase the risk of thrombosis by inhibiting and lowering tissue plasminogen activators (tPA, a fibrinolysis marker) [42]. In cell models, 1,25(OH)2D decreases the production and activity of PAI-1 [43]. In type 2 diabetes mellitus patients with vitamin D deficiency (<20 ng/mL), a baseline serum 25(OH)D was inversely associated with PAI-1 levels, and a 180-day intervention of 2000 IU of oral vitamin D (cholecalciferol) daily lead to an increase in vitamin D levels and a decrease in PAI-1 levels significantly [44]. Therefore, vitamin D deficiency may impair the balance between PAI-1 and tPA to increase the risk of thrombosis. Taken together, these molecular mechanisms and clinical findings provide support for the link between a low vitamin D status and an increased risk of VTE.

The finding of our subgroup analysis that demonstrated a non-significant impact of study design (i.e., case-control vs. non-case-control) on the risk of VTE was in conflict with the stratified result of a previous meta-analysis [21]. In addition to the relatively small sample size (602 cases) of the case-control studies included in that study, possible dilution of the effect of vitamin D deficiency by setting a single threshold of low vitamin D at 30 ng/mL in that study for analysis without differentiating between those with vitamin D insufficiency (i.e., 20–30 ng/mL) and deficiency (i.e., <20 ng/mL) as in our study may have biased their findings. Although it was noted that patients with neurological diseases are at high risk of VTE because of relative immobility and a hypercoagulable state [22], our subgroup analysis showed that low vitamin D status was linked to an increased risk of VTE regardless of the presence of neurological diseases. A discrepancy in the numbers for comparison (n = 812 vs. 7021 for those with vs. without neurological diseases, respectively), as well as a lack of information about the timing of onset of neurological disease and follow-up duration, may contribute to our negative finding.

Regarding the causal relationship between vitamin D deficiency and the risk of VTE, there were two previous randomized controlled studies attempting to address the issue [45,46]. One large-scale randomized, double-blind, placebo-controlled trial (RCT) involving 36,282 postmenopausal women aged 50–79 years in the United States demonstrated no significant effect of the supplementation with 400 IU of vitamin D and 1000 mg of calcium on the overall risk of VTE after an average follow-up of up to seven years [45]. In that RCT study, a VTE was classified as idiopathic or secondary. A VTE was defined as secondary if procedure-related, within 3 months of a fracture or an inpatient hospitalization, with a history of cancer, or with current use of oral hormone therapy. All other VTE events were considered idiopathic. The secondary analyses of that study revealed a lower risk of idiopathic VTE in women randomized to receive vitamin D and calcium [45]. Furthermore, subgroup analyses in that study suggested a protective effect of vitamin D and calcium supplementation against idiopathic VTE among women with low baseline 25(OH)D concentrations [45]. Another clinical study recruiting obese and overweight subjects reported significant associations of their baseline serum 25(OH)D concentrations with the circulating tPA and PAI-1 levels [46]. However, the significance of such correlations disappeared after a one-year intervention with high-dose oral vitamin D supplementation, thereby adding more uncertainties to this issue [46]. Furthermore, a case-controlled study using data from the Multiple Environmental and Genetic Assessment database between 1999 and 2004 showed no significant correlation between vitamin D supplementation and a decreased risk of VTE [47]. In contrast, a retrospective cohort study in a rehabilitation center in the U.S. focusing on acute spinal cord injury patients with low vitamin D levels (<30 ng/mL) reported a reduced risk of VTE in those with vitamin D supplementation compared to those without [31]. Therefore, the association between vitamin D supplementation and the risk of VTE remains inconsistent. Future studies are warranted not only to elucidate a possible causal association between vitamin D status and the risk of VTE but also to identify the role of vitamin D supplementation in the reduction of the risk of VTE.

Compared with the previous meta-analysis [21], our study had several advantages. First, our meta-analysis included 16 observational studies with 47,648 individuals to provide more robust evidence on this topic. Second, the previous meta-analysis investigated the association of vitamin D status with the risk of VTE by combining OR and HR data. In contrast, we analyzed the data on OR and HR separately to minimize possible bias. Third, we investigated the dose-dependent correlation between vitamin D levels and the risk of VTE, which was not addressed in the previous meta-analysis. Fourth, our meta-analysis included studies in different clinical settings so that our findings may be applied to a wide spectrum of individuals, from the general population to those in various disease statuses.

Regarding specific individuals who may be particularly exposed to an increased risk of VTE associated with a low vitamin D level, a previous investigation highlighted an ethnic impact on the risk of VTE [3]. Moreover, underlying diseases or the type of surgery may be linked to an elevated risk of VTE. Of the 16 studies included in the current meta-analysis, three recruiting patients with vitamin D deficiency (i.e., a circulating level <20 ng/mL) showed that hospitalized patients with COVID-19 [37], those receiving total knee arthroplasty [33], and those undergoing hip fracture surgery [34] might be more susceptible to the vitamin D deficiency-related increase in the risk of VTE compared to the other patient populations. The hyperinflammatory condition of COVID-19 may result in a low vitamin D level [37], which may exacerbate their hypercoagulative status. Future studies are needed to elucidate the effect of vitamin D deficiency on the risk of VTE in these patient populations.

There were several limitations in the present meta-analysis. First, our analysis of retrospectively collected and limited data on outcomes from previous reports precluded our investigation into the association of 25(OH)D levels with the severity of VTE. Second, this study did not take into account the seasonal effects on 25(OH)D levels, given the finding of the lowest vitamin D concentration in winter [48]. Third, the known significant ethnic variation in the incidence of VTE [49] may not justify the extrapolation of our results to populations of different ethnic backgrounds. Fourth, despite the gender difference in the prevalence of VTE and vitamin D levels [38], this association was not analyzed due to a lack of relevant data. Fifth, we did not analyze other known confounders related to blood coagulation, such as the presence of active cancer or coagulopathy, which may influence the development of DVT.

In conclusion, this meta-analysis based on observational studies demonstrated that vitamin D deficiency (i.e., a circulating level less than 20 ng/mL) may be associated with an increased risk of VTE. Subgroup analysis revealed that the outcome was affected neither by the study designs (e.g., case-control studies vs. non-case-control studies) nor the presence of neurological diseases. Given the modifiable nature of a low circulating vitamin D status, normalization of circulating vitamin D levels with oral supplementation may be beneficial in the reduction of the risk of VTE. Further studies are needed to verify the role of vitamin D supplementation in the development of VTE.

## Figures and Tables

**Figure 1 nutrients-15-01113-f001:**
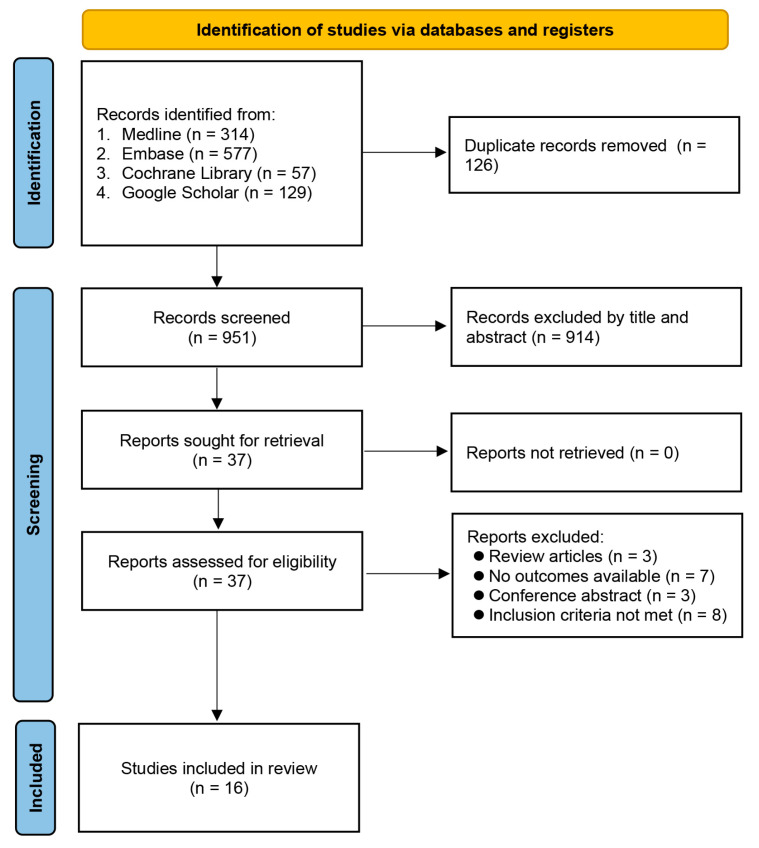
PRISMA flow diagram of study selection.

**Figure 2 nutrients-15-01113-f002:**
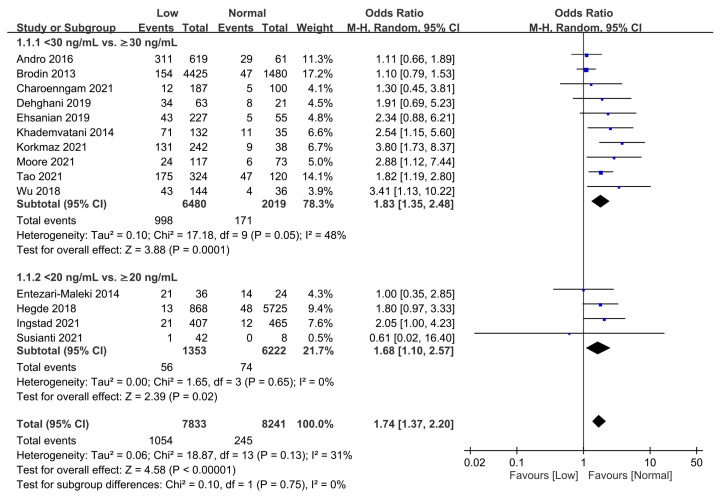
Forest plot comparing the overall risk of venous thromboembolism estimated with odds ratio between individuals with and those without a low vitamin D status. CI, confidence interval; M-H, Mantel-Haenszel.The measure of effect (e.g., an odds ratio) for each of these studies is represented by a square incorporating confidence intervals represented by horizontal lines. The vertical line is “Line of no effect”. The area of each square is proportional to the study’s weight in the meta-analysis. This meta-analysed measure of effect is plotted as a diamond, the lateral points of which indicate confidence intervals for this estimate [18,19,28,29,30,31,32,33,34,35,36,37,38,39].

**Figure 3 nutrients-15-01113-f003:**
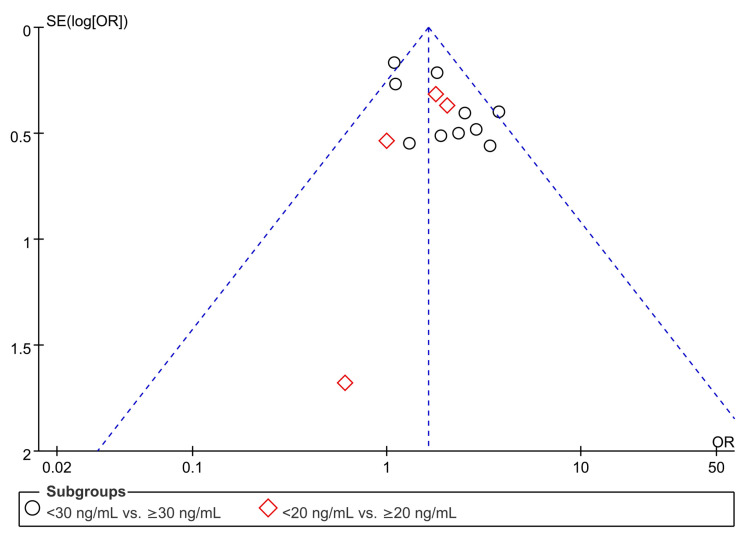
Funnel plot showing a low risk of publication bias.

**Figure 4 nutrients-15-01113-f004:**
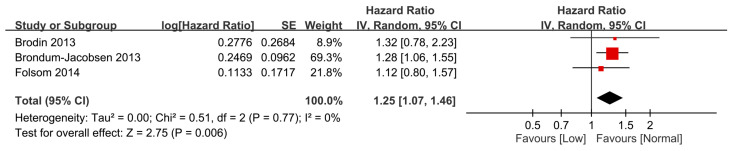
Forest plot comparing the overall risk of venous thromboembolism based on a time-to-event analysis between participants with and without a low vitamin D status. CI, confidence interval; SE, standard error.The measure of effect (e.g., an odds ratio) for each of these studies is represented by a square incorporating confidence intervals represented by horizontal lines. The vertical line is “Line of no effect”. The area of each square is proportional to the study’s weight in the meta-analysis. This meta-analysed measure of effect is plotted as a diamond, the lateral points of which indicate confidence intervals for this estimate [7,19,20].

**Figure 5 nutrients-15-01113-f005:**
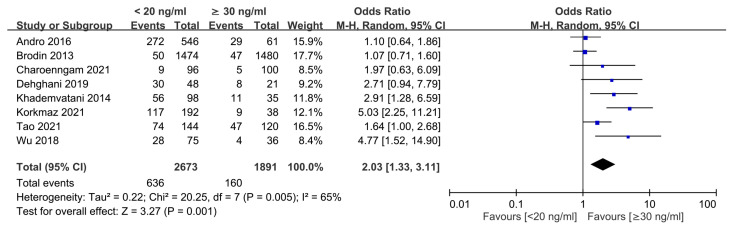
Subgroup analysis comparing the risk of venous thromboembolism between individuals with vitamin D deficiency and those with a normal vitamin D status. CI, confidence interval; M-H, Mantel-Haenszel. The measure of effect (e.g., an odds ratio) for each of these studies is represented by a square incorporating confidence intervals represented by horizontal lines. The vertical line is “Line of no effect”. The area of each square is proportional to the study’s weight in the meta-analysis. This meta-analysed measure of effect is plotted as a diamond, the lateral points of which indicate confidence intervals for this estimate [18,19,28,29,30,35,38,39].

**Figure 6 nutrients-15-01113-f006:**
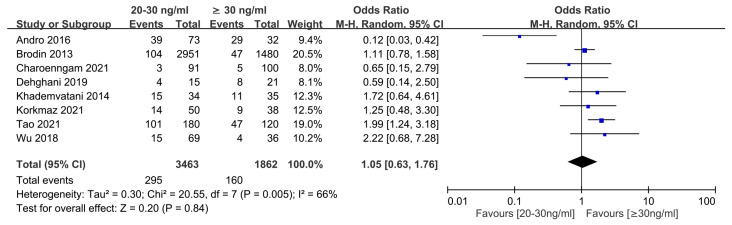
Subgroup analysis comparing the risk of venous thromboembolism between participants with vitamin D insufficiency and those with a normal vitamin D status. CI, confidence interval; M-H, Mantel-Haenszel. The measure of effect (e.g., an odds ratio) for each of these studies is represented by a square incorporating confidence intervals represented by horizontal lines. The vertical line is “Line of no effect”. The area of each square is proportional to the study’s weight in the meta-analysis. This meta-analysed measure of effect is plotted as a diamond, the lateral points of which indicate confidence intervals for this estimate [18,19,28,29,30,35,38,39].

**Figure 7 nutrients-15-01113-f007:**
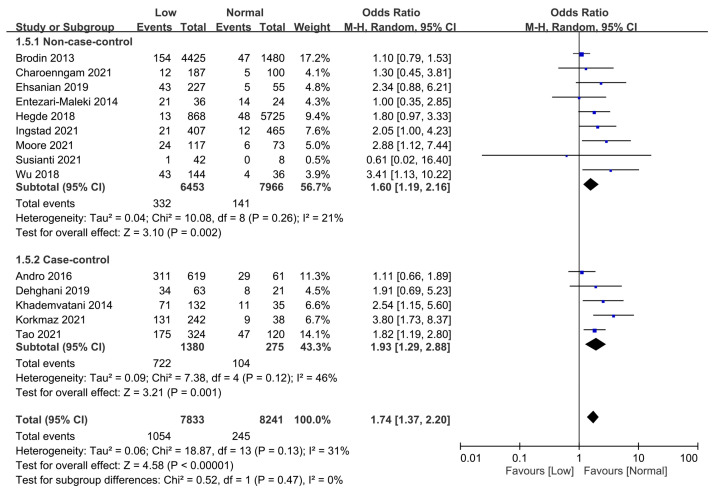
Subgroup analysis focusing on the influence of study design (i.e., non-case-control vs. case-control) on the association between vitamin D status and the risk of venous thromboembolism. CI, confidence interval; M-H, Mantel-Haenszel. The measure of effect (e.g., an odds ratio) for each of these studies is represented by a square incorporating confidence intervals represented by horizontal lines. The vertical line is “Line of no effect”. The area of each square is proportional to the study’s weight in the meta-analysis. This meta-analysed measure of effect is plotted as a diamond, the lateral points of which indicate confidence intervals for this estimate [18,19,28,29,30,31,32,33,35,36,37,38,39].

**Figure 8 nutrients-15-01113-f008:**
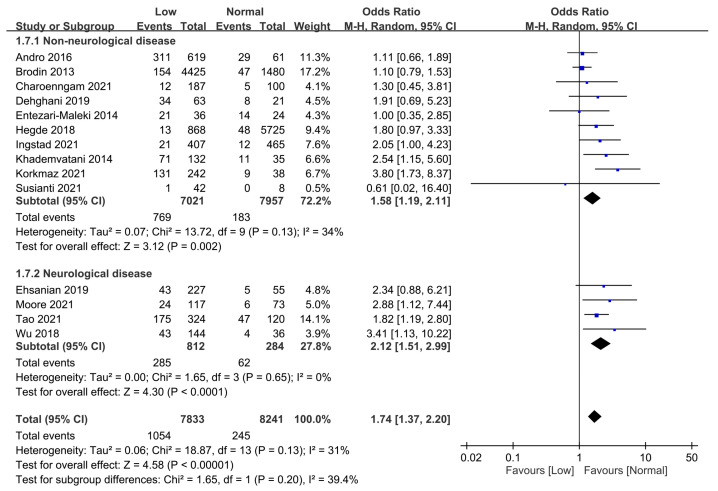
Subgroup analysis on the effect of the presence of neurological diseases (i.e., with vs. without neurological diseases) on the correlation between serum vitamin D level and the risk of venous thromboembolism. CI, confidence interval; M-H, Mantel-Haenszel. The measure of effect (e.g., an odds ratio) for each of these studies is represented by a square incorporating confidence intervals represented by horizontal lines. The vertical line is “Line of no effect”. The area of each square is proportional to the study’s weight in the meta-analysis. This meta-analysed measure of effect is plotted as a diamond, the lateral points of which indicate confidence intervals for this estimate [18,19,28,29,30,31,32,33,34,35,36,37,38,39].

**Table 1 nutrients-15-01113-t001:** Characteristics of included studies (n = 16).

First Author (Years)	Country	Mean Age (Years)	Male (%)	Population	Sample Size	Study Design	NOS	Vitamin D Threshold (ng/mL)	Measurements of 25-Hydroxyvitamin D	Data for Analysis
Brøndum-Jacobsen (2013) [7]	Denmark	57	45	GP	18791	Prospective cohort	6	30	Plasma—CLIA	HR
Brodin (2013) [19]	Norway	62	46	GP	6021	Prospective cohort	6	28	Serum—ECLIA	OR HR
Folsom (2014) [20]	U.S.A.	56.8	24–51	GP	12752	Prospective cohort	6	18	Serum—LC-MS/MS	HR
Entezari-Maleki (2014) [32]	Iran	54.7	53.3	DVT or PE	60	Prospective cross-section	8	20	Plasma—RIA	OR
Khademvatani (2014) [18]	Iran	47.1	48.5	Idiopathic LE DVT vs. HC	167	Case-control	9	30	Plasma—CLIA	OR
Andro (2016) [28]	France	81.7	31.5	DVT or PE vs. HoC	680	Case-control	7	30	Serum—CLIA	OR
Wu (2018) [39]	China	68 vs. 63 ^†^	36	Ischemic stroke	180	Prospective cross-section	8	30	Plasma—CLIA	OR
Hegde (2018) [33]	U.S.A.	65–69 vs. 70–74 ^‡^	24.3	TKA	6593	Retrospective cohort	8	20	Plasma—Procedure not available	OR
Dehghani (2019) [30]	Iran	47.5	61.9	LE DVT or PE vs. HC	84	Case-control	9	30	Plasma—CLIA	OR
Ehsanian (2019) [31]	Canada	45	72	SCI	282	Retrospective cohort	7	30	Serum—LC-MS/MS	OR
Charoenngam (2021) [29]	U.S.A.	62	52.6	COVID-19 infection	287	Retrospective cross-section	8	30	Serum—CLIA	OR
Ingstad (2021) [34]	Norway	81	34	Hip fracture	872	Retrospective cohort	9	20	Serum—LC-MS/MS	OR
Korkmaz (2021) [35]	Turkey	60	49.3	LE DVT vs. HC	280	Case-control	9	30	Plasma—CLIA	OR
Moore (2021) [36]	U.S.A.	65	66	TBI	190	Retrospective cohort	8	30	Serum—ECLIA	OR
Susianti (2021) [37]	Indonesia	53 vs. 59	54	COVID-19 infection	50	Prospective cross-section	9	20	Serum—ELISA	OR
Tao (2021) [38]	China	70.8	59	Ischemic stroke	444	Case-control	9	30	Serum—ECLIA	OR

DVT: deep vein thrombosis; GP: general population; HC: healthy control; HoC: hospitalized control; LE: lower-extremity; PE: pulmonary embolism; TBI: traumatic brain injury; TKA: total knee arthroplasty; SCI: spinal cord injury. CLIA: Chemiluminescence immunoassay; ECLIA: Electro-chemiluminescence immunoassay; ELISA: Enzyme-linked immunosorbent assay; LC-MS/MS: Liquid chromatography in tandem with mass spectrometry; RIA: Radio-immunoassay; ^†^ DVT group vs. non DVT group. ^‡^ Vitamin D deficient group vs. vitamin D sufficient group. Low vitamin D group vs. normal vitamin D group; NOS: Newcastle-Ottawa Scale; OR: odds ratio; HR: hazard ratio.

## Data Availability

Not applicable.

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
