# Peer review of "Is Circulating Vitamin D Status Associated with the Risk of Venous Thromboembolism? A Meta-Analysis of Observational Studies"

_nutrients, 2023, doi:10.3390/nu15051113_

Round 1

Reviewer 1 Report

Dear Editor,

the present meta-analysis investigated the impact of low vitamin D levels on the risk for venous thromboembolism (VTE). Authors concluded that 25(OH)D lower than 20 ng/ml was associated with increased risk of VTE independently from the presence of neurological disorder. Although the evidence was only supported  by observational studies, the analysis looks methodically conducted and results were clearly presented. Authors purposed a well referenced rationale of anti-thrombotic effect elicited by the vitamin D system.

Few minor considerations could be raised prior to publication.

- The paucity of randomized controlled trials, having tested efficacy of vitamin D administration against the incidence of VTE, hardly limits encouraging preventive vitamin D replenishment. Furthermore, in the present metanalysis only 25(OH)D levels < 20 ng/ml were associated with the risk for VTE. Authors may soften conclusions and comment on regard.

- In the Authors opinion, which populations could me more exposed to the impact of vitamin D deficiency on VTE?

Author Response

General comment:

The present meta-analysis investigated the impact of low vitamin D levels on the risk for venous thromboembolism (VTE). Authors concluded that 25(OH)D lower than 20 ng/ml was associated with increased risk of VTE independently from the presence of neurological disorder. Although the evidence was only supported by observational studies, the analysis looks methodically conducted and results were clearly presented. Authors purposed a well referenced rationale of anti-thrombotic effect elicited by the vitamin D system.

Response:

We would like to express our gratitude to the Reviewer for the time taken to review our manuscript and the professional comments that substantially improved the quality of our work. Please find our point-by-point response below. Please also kindly note that the corresponding changes made in the revised manuscript are marked in blue.

Few minor considerations could be raised prior to publication.

Comment 1:

The paucity of randomized controlled trials, having tested efficacy of vitamin D administration against the incidence of VTE, hardly limits encouraging preventive vitamin D replenishment. Furthermore, in the present metanalysis only 25(OH)D levels < 20 ng/ml were associated with the risk for VTE. Authors may soften conclusions and comment on regard.

Response 1:

We are grateful to the Reviewer for the highly insightful comment. We have added the following information in the Discussion section (fourth paragraph) of the revised manuscript and modified Conclusion (marked in blue).

“Furthermore, a case-controlled study using data from the Multiple Environmental and Genetic Assessment database between 1999 and 2004 showed no significant correlation between vitamin D supplementation and a decreased risk of VTE (Vučković et al.,2015). In contrast, a retrospective cohort study in a Rehabilitation Center in the U.S. focusing on acute spinal cord injury patients with low vitamin D levels (<30 ng/mL) reported a reduced risk of VTE in those with vitamin D supplementation compared to those without (Ehsanian et al., 2019). Therefore, the association between vitamin D supplementation and the risk of VTE remains inconsistent.  Future studies are warranted not only to elucidate a possible causal association between vitamin D status and the risk of VTE but also to identify the role of vitamin D supplementation in the reduction of the risk of VTE.

References

Vučković BA, van Rein N, Cannegieter SC, Rosendaal FR, Lijfering WM. Vitamin supplementation on the risk of venous thrombosis: results from the MEGA case-control study. Am J Clin Nutr. 2015 Mar;101(3):606-12.

Ehsanian R, Timmerman MA, Wright JM, McKenna S, Dirlikov B, Crew J. Venous Thromboembolism is Associated With Lack of Vitamin D Supplementation in Patients With Spinal Cord Injury and Low Vitamin D Levels. PM R. 2019 Feb;11(2):125-134.

Comment 2:

In the Authors opinion, which populations could me more exposed to the impact of vitamin D deficiency on VTE?

Response 2:

We are grateful to the Reviewer for the professional inquiry, in response to which we have added relevant descriptions with appropriate references into the Discussion section (sixth paragraph) of the revised manuscript to provide this information for the readers.

“Regarding specific individuals who may be particularly exposed to the risk of VTE associated with a low vitamin D level, a previous investigation has highlighted an ethnic impact on the risk of VTE (White et al, 2005). Besides, underlying diseases or the type of surgery may be linked to an increased risk of VTE. Of the 16 studies included in the current meta-analysis, three recruiting patients with vitamin D deficiency (i.e., a circulating level < 20 mg/mL) showed that hospitalized patients with COVID-19 (Susianti et al, 2021), those receiving total knee arthroplasty (Hegde et al, 2018), and those undergoing hip fracture surgery (Ingstad et al, 2021) may be more susceptible to the vitamin D deficiency-related increase in risk of VTE compared to the other patient populations. The hyperinflammatory condition in COVID-19 may result in a low vitamin D level (Susianti et al, 2021), which may exacerbate their hypercoagulative status. Future studies are needed to elucidate the effect of vitamin D deficiency on the risk of VTE in these patient populations.”

References

  1. White RH, Zhou H, Murin S, Harvey D. Effect of ethnicity and gender on the incidence of venous thromboembolism in a diverse population in California in 1996. Thromb Haemost. 2005 Feb;93(2):298-305.

  1. Susianti H, Wahono CS, Rahman PA, Pratama MZ, Wulanda IA, Hartanti KD, Dewi ES, Handono K. Low levels of vitamin D were associated with coagulopathy among hospitalized coronavirus disease-19 (COVID-19) patients: A single-centered study in Indonesia. J Med Biochem. 2021 Sep 3;40(4):341-350.

  1. Hegde V, Arshi A, Wang C, Buser Z, Wang JC, Jensen AR, Adams JS, Zeegen EN, Bernthal NM. Preoperative Vitamin D Deficiency Is Associated With Higher Postoperative Complication Rates in Total Knee Arthroplasty. Orthopedics. 2018 Jul 1;41(4):e489-e495.

  1. Ingstad F, Solberg LB, Nordsletten L, Thorsby PM, Hestnes I, Frihagen F. Vitamin D status and complications, readmissions, and mortality after hip fracture. Osteoporos Int. 2021 May;32(5):873-881.

Reviewer 2 Report

Hung et al. performed a meta-snalysis on the relationship between  serum/plasma levels of vitamin D and the risk of VTE based on data from16 pulished studies.  In the past, the group has published a number of similar type of correlative studies on different diseases using the same approach.  This study is solely based on statistical analyses of data from different studies.  In this regard, it is critical to establish that appropriate statistical methods were employed.  Please see specific comments listed below.

1. As stated above, it is critical to establish that the correct statistical methods were used for the study.  Without the raw data, it is difficult for an outside statistician to critically assess the validty of this study.  One statistician friend of mine tells me that >90% of biomedical studies use the wrong statistical methods for data analysis.  Recognizing this and other statistics related issues, many institutions in the USA and Europe today require in-house statistician to be involved in any biomedical studies involving statistical approaches.  It is my strong recommendation that the authors recruite a statistician to go over the study to confirm that the study was done correctly, especcially when combining data from different studies.

2.  This paper is submitted as a review.  Goingn over the manuscript, I feel that the paper should be called an article.  In fact, the style of presentation is that of an article, not a review.

3.  The authors discuss dose-dependency.  I am not sure if this is a critical issue, especially from the point of view of clinical application/treatment of VTE patients.  Lack of dose-dependency seems reasonable since the normal range of vitamin D concentration in serum is wide (in many cases anything above 20 ng/ml up to 40-50 ng/ml).  The study seems to say that vitamin D deficiency but not insufficiency is associated with VTE. The border of the two conditions is around 20 ng/ml.  Indeed, many clinicians seem to recommend keeping the serum vitamin D concentration above 20 ng/ml.  I suggest toning down the emphasis on dose-dependency, which is not a critical issue.

There are several minor points.

i.  The backgroud section of Abstract.  The description given has nothing to do the the background of the study.  Describe the rational for doing the study with appropriate known facts which support the need of this study.

ii.  Lines 45 - 47.  There are two sentences here.  The first sentence states that there are substantial differences in VTE incidence among different ethnic groups.  However, in the second sentence which gives specific examples, the authors weaken their claim by saying that there may be a difference.  This is not confusing.  Is there a difference or not?  What does statistics say?

iii.  Line 57.  Delete "in endothelial cells."

iv.  Line 79.  Define PROSPERO.

v.  Line 91.  There is no Appendix 2.  Where is Appendix 1?

vi.  Results.  References are not cited in the numerical sequence.  Why?

vii.  Table 1 legend.  Why is the full description of ECLIA in italic?

viii.  Discussion, lines 311 - 326.  This discussion is not critical.  Suggest deletion.

ix.  Lines 343 - 346.  This sentence is imcomplete.  Why "while"?

x.  Lines 383 - 385.  Statements on insufficiency is not critical unless the authors provide some strong rationals for studying this aspect.

Author Response

General comment:

Hung et al. performed a meta-snalysis on the relationship between serum/plasma levels of vitamin D and the risk of VTE based on data from16 pulished studies.  In the past, the group has published a number of similar type of correlative studies on different diseases using the same approach.  This study is solely based on statistical analyses of data from different studies.  In this regard, it is critical to establish that appropriate statistical methods were employed.  Please see specific comments listed below.

Response:

We would like to take this opportunity to express our gratitude to the Reviewer for taking valuable time to review our manuscript. In compliance with the professional suggestions of the Reviewer, we have revised our manuscript accordingly. Please find below our point-by-point responses. Please also kindly note that the corresponding changes in the revised manuscript are marked in red.

Comment 1:

As stated above, it is critical to establish that the correct statistical methods were used for the study.  Without the raw data, it is difficult for an outside statistician to critically assess the validty of this study.  One statistician friend of mine tells me that >90% of biomedical studies use the wrong statistical methods for data analysis.  Recognizing this and other statistics related issues, many institutions in the USA and Europe today require in-house statistician to be involved in any biomedical studies involving statistical approaches.  It is my strong recommendation that the authors recruite a statistician to go over the study to confirm that the study was done correctly, especcially when combining data from different studies.

Response 1:

In compliance with the Reviewer’s insightful suggestion, we have consulted a professional statistician of the Department of Public Health, Institute of Epidemiology and Preventive Medicine, College of Public Health, National Taiwan University, Taipei, Taiwan, Prof. Yu-Kang Tu, who noticed that the studies we included were all observational studies, for which he suggested additional data after adjustment of the confounding factors (e.g., the adjusted odds ratios) to serve as a reference for the readers. Therefore, after reviewing the 16 included studies, four provided the adjusted odds ratio for analyzing the impact of a low vitamin level on the risk of VTE. In response to the Reviewer’s professional comment, we have added relevant results into the Result section (3.2.1.1 Association of low vitamin D with the risk of VTE estimated with odds ratio) of the revised manuscript:

“Meta-analysis of the four studies that provided the adjusted OR for analyzing the impact of a low vitamin D level on the risk of VTE showed a positive association between the former and the latter (adjust OR: 1.71, 95% CI:1.24 to 2.38, p=0.001, I2=45%, 7503 patients) (Figure not shown).”

Comment 2:

This paper is submitted as a review.  Going on over the manuscript, I feel that the paper should be called an article.  In fact, the style of presentation is that of an article, not a review.

Response 2:

We are grateful to the Reviewer for the professional comment, in compliance with which we have changed the submission category from “Review” to “Article”.

Comment 3:

The authors discuss dose-dependency.  I am not sure if this is a critical issue, especially from the point of view of clinical application/treatment of VTE patients.  Lack of dose-dependency seems reasonable since the normal range of vitamin D concentration in serum is wide (in many cases anything above 20 ng/ml up to 40-50 ng/ml).  The study seems to say that vitamin D deficiency but not insufficiency is associated with VTE. The border of the two conditions is around 20 ng/ml.  Indeed, many clinicians seem to recommend keeping the serum vitamin D concentration above 20 ng/ml.  I suggest toning down the emphasis on dose-dependency, which is not a critical issue.

Response 3:

The Reviewer’s insightful suggestion is sincerely appreciated. Accordingly, we have deleted this part of the relevant description in the Discussion section.

“To the best of our knowledge, the current meta-analysis is the first to demonstrate a dose-dependent effect of vitamin D levels on the risk of VTE. Similarly, a dose-dependent effect of preoperative vitamin D levels on the risk of postoperative delirium was found in our previous meta-analysis [11]. A cross-sectional nationwide population-based survey revealed a significant dose-dependent trend between higher serum vitamin D levels and a lower risk of hypertension in pre-menopausal women after adjusting for sociodemographic, behavioral, and dietary factors [44]. The finding was also supported by previous experimental studies. An in vitro study has identified not only elevated levels of PAI-1 and thrombospondin-1 as risk factors of thrombosis, but also a dose-dependent relationship between calcitriol and the down-regulation of PAI-1 and thrombospondin-1 in aortic smooth muscle cells [45]. Moreover, in human periodontal ligament fibroblasts and primary periodontal ligament cells, both of 25(OH)D and 1,25(OH)2D significantly inhibit proinflammatory mediators including IL-6, IL-8 and monocyte chemotactic protein-1 in a dose-dependent manner [46]. Taken together, the findings of previous experimental and clinical human studies that demonstrated the dose-dependent effects of vitamin D on multiple pathological conditions supported our result in the VTE setting.”

There are several minor points.

Comment 4: 

The background section of Abstract.  The description given has nothing to do the the background of the study.  Describe the rational for doing the study with appropriate known facts which support the need of this study.

Response 4:

We would like to thank the Reviewer for the professional suggestion, in compliance with which we have rewritten the background section of the Abstract.

“Background: Although vitamin D is theoretically anti-thrombotic, associations between serum vitamin D status and the risk of venous thromboembolism (VTE) remain inconsistent”

Comment 5:

Lines 45 - 47.  There are two sentences here.  The first sentence states that there are substantial differences in VTE incidence among different ethnic groups.  However, in the second sentence which gives specific examples, the authors weaken their claim by saying that there may be a difference.  This is not confusing.  Is there a difference or not?  What does statistics say?

Response 5:

The Reviewer’s insightful concern is highly appreciated. We have rewritten the description in the Introduction section (first paragraph) of the revised manuscript with the addition of relevant data and the citation of an appropriate reference.

“The incidence of VTE varies substantially among different ethnic groups [White et al, 2005]. A cohort study using the California Discharge Data in 1996 revealed that the incidence of VTE per 100,000 was 104 in Caucasians, 141 in African-Americans, 55 in Hispanics, and 21 in Asians, indicating a higher incidence of VTE in African-Americans compared with a relatively low incidence in Asians [White et al, 2005].”

Reference

White RH, Zhou H, Murin S, Harvey D. Effect of ethnicity and gender on the incidence of venous thromboembolism in a diverse population in California in 1996. Thromb Haemost. 2005 Feb;93(2):298-305.

Comment 6:

Line 57.  Delete "in endothelial cells."

Response 6:

We are truly thankful to the Reviewer for the professional suggestion, in response to which we have deleted "in endothelial cells."

Comment 7:

Line 79.  Define PROSPERO.

Response 7:

In compliance with the Reviewer’s insightful comment, we have added the full term of PROSPERO (i.e., The International Prospective Register of Systematic Reviews) in the Method section of the revised manuscript.

“The current meta-analysis was conducted in accordance with the Preferred Reporting Items for Systematic Reviews and Meta‐analyses (PRISMA) checklist [22] and registered in the International Prospective Register of Systematic Reviews (PROSPERO, https://www.crd.york.ac.uk/prospero/, registration no.: CRD42021228606).”

Comment 8:

Line 91.  There is no Appendix 2.  Where is Appendix 1?

Response 8:

We would like to thank the Reviewer for the insightful reminder. In fact, there was only one Appendix. The mistake has been corrected accordingly.

Comment 9:

Results.  References are not cited in the numerical sequence.  Why?

Response 9:

The Reviewer’s professional comments are highly appreciated. In the main text, we cited the included studies according to the numerical order in which they appeared in the manuscript starting from the Introduction. In the figures (i.e., forest plots), we listed the included studies in alphabetical order to facilitate the identification of a specific reference by the readers.

Comment 10:

Table 1 legend.  Why is the full description of ECLIA in italic?

Response 10:

We would like to apologize for the mistake and are thankful to the Reviewer for the meticulous observation. We have changed the font style of the full description of ECLIA from italic to "regular".

Comment 11:

Discussion, lines 311 - 326.  This discussion is not critical.  Suggest deletion.

Response 11:

In compliance with the Reviewer’s professional suggestion, we have deleted the description from the Discussion section of the revised manuscript.

“To the best of our knowledge, the current meta-analysis is the first to demonstrate a dose-dependent effect of vitamin D levels on the risk of VTE. Similarly, a dose-dependent effect of preoperative vitamin D levels on the risk of postoperative delirium was found in our previous meta-analysis [11]. A cross-sectional nationwide population-based survey revealed a significant dose-dependent trend between higher serum vitamin D levels and a lower risk of hypertension in pre-menopausal women after adjusting for sociodemographic, behavioral, and dietary factors [44]. The finding was also supported by previous experimental studies. An in vitro study has identified not only elevated levels of PAI-1 and thrombospondin-1 as risk factors of thrombosis, but also a dose-dependent relationship between calcitriol and the down-regulation of PAI-1 and thrombospondin-1 in aortic smooth muscle cells [45]. Moreover, in human periodontal ligament fibroblasts and primary periodontal ligament cells, both of 25(OH)D and 1,25(OH)2D significantly inhibit proinflammatory mediators including IL-6, IL-8 and monocyte chemotactic protein-1 in a dose-dependent manner [46]. Taken together, the findings of previous experimental and clinical human studies that demonstrated the dose-dependent effects of vitamin D on multiple pathological conditions supported our result in the VTE setting.”

Comment 12:

Lines 343 - 346.  This sentence is incomplete.  Why "while"?

Response 12:

Thanks for the suggestion. We have deleted the word “While” from the sentence.

Comment 13: 

Lines 383 - 385.  Statements on insufficiency is not critical unless the authors provide some strong rationals for studying this aspect.

Response 13:

In response to the Reviewer’s professional suggestion, we have rewritten the relevant descriptions in the Discussion and conclusion sections and deleted the original statements on insufficiency.

“This meta-analysis demonstrated that vitamin D deficiency may be linked to an increased risk of VTE. Subgroup analysis further revealed no significant impact of study designs (e.g., case-control vs. non-case-control) and the presence of neurological diseases on the risk of VTE.” (Discussion section, first paragraph)

“In conclusion, this meta-analysis demonstrated that vitamin D deficiency (i.e., a circulating level less than 20 ng/mL) may be associated with an increased risk of VTE.” (Conclusion sections)
